# Large Language Models as Interfaces to Structured Data: A Survey

## Abstract

Structured data, including tables, relational databases, and knowledge graphs, underpins a wide range of scientific, industrial, and decision-making workflows. Although large language models (LLMs) are primarily trained on unstructured text, recent work has demonstrated their effectiveness in tasks involving structured data, such as table reasoning, natural language to SQL translation, data transformation, and automated analytics. These developments indicate that LLMs can function as a general interface between natural language inputs, structured representations, and executable operations.

This survey presents a theory-oriented overview of LLMs for structured data. We introduce an abstract formulation that characterizes structured data tasks by the structured state, the query or control input, the output space, and the execution environment. Based on this formulation, which we revisit throughout the taxonomy and evaluation sections, we propose a taxonomy that organizes existing methods according to the functional role of the LLM, including encoding, reasoning, translation, planning, and agent-based execution, as well as by representation strategies and learning signals. This taxonomy highlights shared design principles across different task settings and clarifies methodological trade-offs.

We examine evaluation protocols, generalization properties, and failure modes specific to structured data tasks, with an emphasis on faithfulness, schema robustness, and execution correctness. Finally, we outline open research directions for LLM-based structured data systems, including challenges related to scalability, symbolic and neural integration, and learning with execution-based supervision. The survey aims to provide a unified conceptual framework and a reference point for future research on large language models applied to structured data.

## 1 Introduction

### 1.1 Motivation: Why Large Language Models for Structured Data

Structured data remains the dominant representation for information in scientific research, enterprise analytics, and data-driven decision-making. Tables, relational databases, and knowledge graphs provide explicit schemas, integrity constraints, and execution semantics that support reliable querying and analysis (Abiteboul et al., 1995; Silberschatz et al., 2019). At the same time, interaction with structured data typically requires specialized query languages or programming expertise, creating barriers for non-expert users and limiting flexibility in exploratory analysis. While natural language to SQL has emerged as a canonical structured data task, a growing body of work applies large language models to structured data settings where no explicit query language is available.

Large language models (LLMs), trained on large-scale text corpora, have demonstrated strong capabilities in natural language understanding, reasoning, and code generation (Brown et al., 2020; Chowdhery et al., 2023). Recent work has shown that these models can be adapted to tasks involving structured data, including table question answering (Liu et al., 2022), natural language to SQL translation (Yu et al., 2018b), data transformation, and automated analytics workflows. In these settings, LLMs enable natural language

interfaces to structured systems while leveraging the formal semantics of underlying execution engines. Figure 1 illustrates this interface-based view, in which large language models mediate between natural language inputs, structured data representations, and external execution environments.

Unifying different structured data paradigms under a shared framework provides several benefits. Although systems for tables, relational databases, and knowledge graphs are often developed in separate research communities, many LLM-based pipelines share similar architectural patterns. For example, tasks across these domains often involve schema understanding, query translation, reasoning over structured representations, and interaction with execution environments. By introducing a common abstraction, the framework allows researchers to identify shared design principles, compare approaches across domains, and transfer ideas between traditionally separate areas such as table question answering and graph querying.

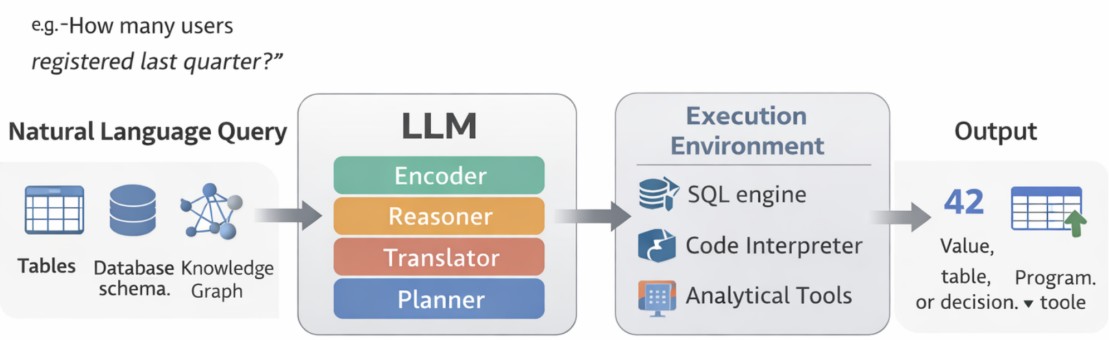

Figure 1: Large language models as interfaces to structured data systems. A natural language query is combined with a structured state, such as tables, database schemas, or knowledge graphs. The large language model operates as an intermediate component performing encoding, reasoning, translation, or planning, while execution is delegated to external environments including SQL engines, interpreters, or analytical tools. The output may take the form of a value, table, program, or decision.

The increasing use of LLMs in structured data contexts reflects a broader shift in how users interact with symbolic systems. Rather than replacing traditional databases or analytical pipelines, LLMs increasingly act as intermediate components that translate, reason, or plan over structured representations while delegating execution to external systems. Understanding this emerging role requires a systematic view that goes beyond individual applications or benchmarks.

## 1.2 Gaps in Existing Surveys

Several surveys have examined related aspects of structured data interaction and language-based modeling. Early surveys focused on semantic parsing and natural language to SQL systems, providing overviews of model architectures, datasets, and evaluation protocols for translating natural language queries into executable programs. Other lines of work surveyed neural models for tabular data, emphasizing representation learning and prediction over structured inputs without explicit language-based interaction (Arık & Pfister, 2021). More broadly, recent surveys of large language models have primarily concentrated on unstructured text and code, treating structured data as a secondary application domain (Bommasani et al., 2021).

More recently, task-specific surveys have emerged that focus on the use of large language models for natural language to SQL translation. These surveys provide comprehensive coverage of model designs, prompting and finetuning strategies, benchmarks, and evaluation trends within the text-to-SQL domain (Shi et al., 2025; Liu et al., 2025b). While these works offer valuable and up-to-date summaries of a canonical structured data task, their scope is intentionally narrow, centering on translation from language to SQL and closely related variations.

Despite this progress, existing surveys do not provide a unified conceptual framework that spans multiple structured data modalities or interaction paradigms. In particular, prior work typically organizes methods by task or model architecture, rather than by the functional role played by the language model within a structured data system. As a result, common methodological patterns across tasks such as table reasoning, data transformation, analytical planning, and agent-based interaction with databases remain fragmented across separate literatures.

Furthermore, existing surveys do not explicitly formalize the interaction between language models, structured representations, and execution environments. The absence of a shared abstraction makes it difficult to reason about design trade-offs related to representation, execution grounding, generalization across schemas, and failure modes that recur across structured data settings.

This survey addresses these gaps by presenting a theory-oriented synthesis of large language models for structured data. Rather than focusing on a single task family, we introduce a unified abstraction for structured data tasks and organize prior work according to the functional role of the language model, including encoding, reasoning, translation, planning, and agentic execution. This perspective enables systematic comparison across diverse applications and clarifies open challenges that cut across structured data modalities.

In contrast to task-specific surveys that focus primarily on natural language to SQL translation, this survey aims to unify a broader class of structured data interaction paradigms under a common abstraction, encompassing tasks with and without explicit query languages, diverse data modalities, and varying degrees of execution grounding.

Table 1: Comparison of this survey with existing surveys on large language models and structured data interaction

| Survey | Year | Primary Scope | Key Limitations |
| --- | --- | --- | --- |
| (Bommasani et al., 2021) | 2021 | Foundation models, capabilities, and societal implications | Broad and unstructured-data centric; does not analyze structured data interaction, execution semantics, or database interfaces. |
| (Lu et al., 2024) | 2024 | LLMs for table understanding and reasoning | Table-centric; does not unify databases, knowledge graphs, or executable structured systems under a shared interaction abstraction. |
| (Fan et al., 2025) | 2025 | Data-centric analysis of datasets, benchmarks, and evaluation practices in NL-to-SQL | Focuses on data quality and benchmark construction; does not provide a functional taxonomy of LLM behaviors or address non-SQL structured data interaction. |
| (Shi et al., 2025) | 2024 | LLM-based Text-to-SQL methods and benchmarks | Restricted to NL-to-SQL translation; does not cover non-SQL structured tasks or agent-based interaction. |
| (Hong et al., 2025) | 2025 | LLM-based Text-to-SQL systems and evaluation | Task-specific focus on SQL generation; lacks a cross-modal taxonomy spanning tables, graphs, and non-query-based structured tasks. |
| **This survey** | 2026 | Unified abstraction and taxonomy of LLMs as interfaces to structured data | Introduces a role-based taxonomy (encoding, reasoning, translation, planning, agents), a formal interaction abstraction, and a comparative analysis of learning signals, execution grounding, and evaluation challenges across structured data modalities. |

Table 1 contrasts this survey with existing reviews on large language models and structured data interaction. Prior work typically focuses on a single task family, most commonly text-to-SQL translation, or treats structured data only as a secondary application of general-purpose language models. In contrast, this survey introduces a unified abstraction and a role-based taxonomy that spans tables, relational databases, and

knowledge graphs, covering both SQL and non-SQL structured settings. This perspective enables systematic comparison across modeling paradigms, learning signals, execution grounding, and evaluation challenges that are not jointly analyzed in existing surveys.

Unlike prior surveys, our goal is not to catalog systems by task alone, but to provide a unifying abstraction that explains how large language models function within structured data pipelines and how design choices recur across domains.

## 1.3 Contributions

The goal of this survey is to provide a conceptual framework and reference point for future research on large language models applied to structured data. Figure 2 provides an overview of the survey organization and highlights how the proposed abstraction, taxonomy, and evaluation discussions are connected across sections. A key contribution of this work is a role-based categorization of LLM behavior in structured data systems, which cuts across task boundaries and reveals shared design patterns that are obscured in task-specific surveys.

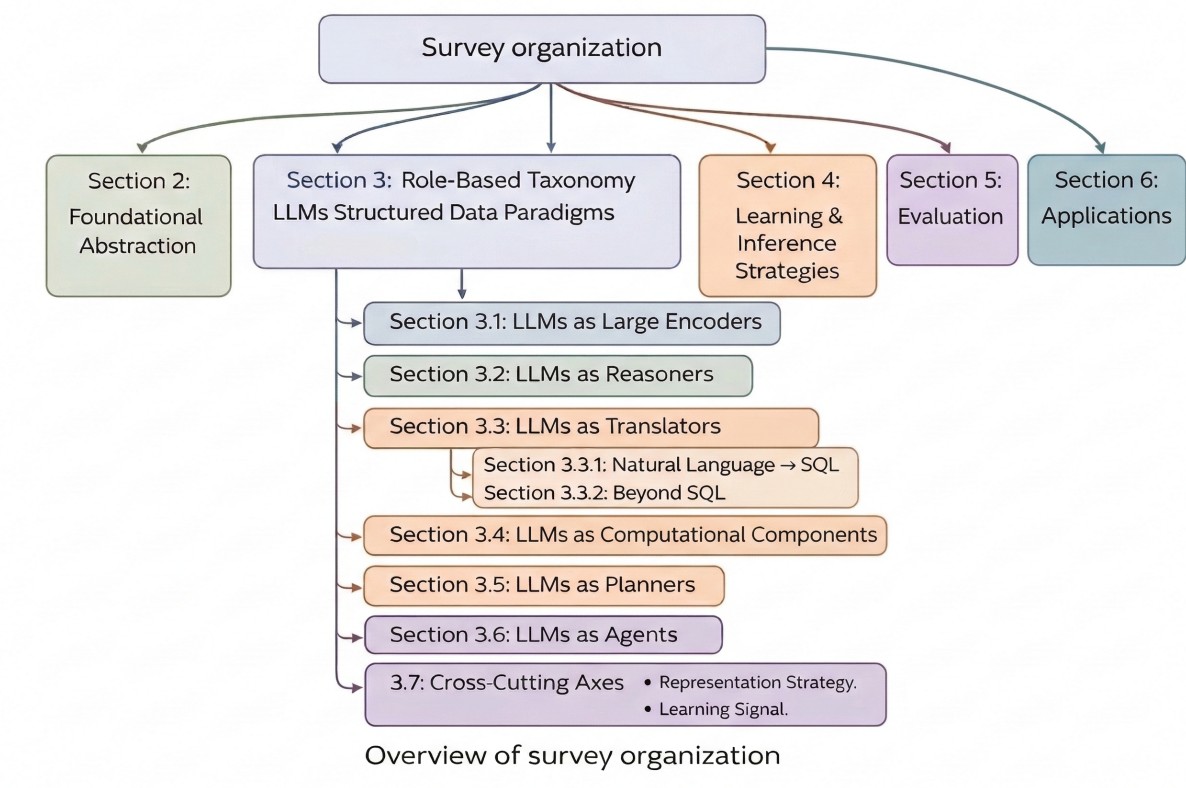

Figure 2: Overview of survey organization. The figure illustrates the logical flow of the paper, beginning with a foundational abstraction of structured data interaction, followed by a taxonomy of large language model paradigms, detailed discussions of translation-based and non-SQL structured tasks, learning and inference strategies, and evaluation considerations.

The main contributions are as follows:

- We introduce a unified abstraction for structured data tasks that formalizes the interaction between structured state, natural language input, execution environment, and outputs.

- We propose a role-based taxonomy that organizes existing approaches by how large language models function within structured data pipelines, including encoding, reasoning, translation, planning, and agentic interaction.

- We synthesize methods across relational databases, tables, and knowledge graphs, showing how shared design patterns emerge across seemingly distinct task families.

- We provide a structured review of benchmarks, evaluation protocols, and failure modes, highlighting limitations of current evaluation practices for structured data interaction.

- We identify open research challenges and future directions related to execution grounding, verification, scalability, and system-level integration.

## 2 Structured Data as a Computational Interface

This section provides the conceptual foundations for the survey. Rather than offering a general background on large language models or structured data systems, we introduce a unifying abstraction that captures how language models interact with structured representations and execution environments. This abstraction serves as the basis for the taxonomy and analysis developed in subsequent sections.

### 2.1 Structured Data Paradigms

Structured data refers to information organized according to an explicit schema and governed by formal semantics. Common instances include relational tables, relational databases, and knowledge graphs. Relational data models represent information as relations with fixed attributes and integrity constraints, enabling declarative querying and optimization through languages such as SQL (Codd, 1970; Silberschatz et al., 2019). Knowledge graphs extend this paradigm by representing entities and relations as labeled graphs, often augmented with ontological constraints and logical rules (Hogan et al., 2021).

These representations support reliable storage, retrieval, and execution by separating logical specification from physical implementation. However, interaction with structured data typically requires familiarity with formal query languages, schema design, and execution semantics. This requirement poses challenges for non-expert users and limits accessibility in exploratory and ad hoc analysis settings.

### 2.2 An Abstract Formulation of Structured Data Tasks

To unify diverse problem settings, we characterize structured data tasks using the following abstract formulation. A structured data task is defined as a tuple

$$(S, Q, O, \mathcal{E}),$$

where:

- $S$ denotes the structured state, such as a table, relational database, or graph, together with its schema;

- $Q$ denotes the query or control input, which may be expressed in natural language, a formal language, or a hybrid representation;

- $O$ denotes the output space, which may include scalar values, structured objects, programs, or decisions;

- $\mathcal{E}$ denotes the execution environment that defines the operational semantics for acting on $S$, such as a database engine, interpreter, or external tool.

Under this formulation, traditional text-to-SQL systems map $(S, Q)$ to a program in $O$ that is executed in $\mathcal{E}$ to produce a result. Table question answering systems may instead map $(S, Q)$ directly to a value in $O$ without exposing an explicit program. Agent-based systems extend this formulation by allowing iterative interaction with $\mathcal{E}$ through multiple intermediate actions (Yao et al., 2023).

### 2.3 From Natural Language Interfaces to LLM-Based Interfaces

Natural language interfaces to structured data have been studied extensively, particularly in the context of semantic parsing and natural language to SQL translation (Zelle & Mooney, 1996; Berant & Liang, 2014). Early approaches relied on rule-based systems or task-specific neural models trained on paired utterance-query data. More recent work introduced large-scale benchmarks such as WikiSQL and Spider, enabling data-driven learning of semantic parsers with increased coverage and complexity (Zhong et al., 2018; Yu et al., 2018b).

Large language models introduce a qualitative change in this landscape. Rather than being trained exclusively for a single structured task, LLMs exhibit broad linguistic and reasoning capabilities that can be adapted to structured data interaction through prompting, finetuning, or tool integration (Brown et al., 2020; Wei et al., 2022). In this setting, the model is not limited to direct translation from language to query, but may perform intermediate reasoning, schema interpretation, or multi-step planning before producing an executable action.

## 3 A Taxonomy of LLM Paradigms for Structured Data

This section presents a taxonomy of existing approaches that apply large language models to structured data tasks. Rather than organizing methods solely by application domain, we categorize prior work according to the functional role played by the LLM within the structured data pipeline. This role-based taxonomy enables comparison across tasks and highlights shared methodological patterns. Table 2 summarizes the primary paradigms for applying large language models to structured data, organized by the functional role played by the model and its interaction with execution environments.

Table 2: Taxonomy of large language model paradigms for structured data

| PARADIGM | LLM ROLE | EXECUTION | REPRESENTATIVE TASKS |
|---|---|---|---|
| Encoder-based | Representation learning over structured inputs | None | Table question answering, fact verification |
| Reasoning-based | Direct reasoning over serialized structures | None | Table-based numerical and logical reasoning |
| Translator-based | Language to executable program mapping | Database or interpreter | Natural language to SQL, data transformation |
| Planner-based | Decomposition into executable steps | Multiple tools | Complex analytical queries, exploratory analysis |
| Agent-based | Iterative interaction and decision making | Environment feedback | Multi-step querying, tool-driven analytics |

Together, these paradigms span the dominant ways in which large language models interact with structured data systems, providing a unifying lens that subsumes task-centric categorizations found in prior work.

### 3.1 LLMs as Encoders of Structured Representations

In the encoder paradigm, the primary role of the LLM is to transform structured data into contextualized representations that can be used for downstream reasoning or prediction. Structured inputs such as tables or schemas are serialized into textual sequences, often using row-wise or column-wise linearization, and processed by the model to produce latent representations (Herzig et al., 2020; Liu et al., 2022).

This paradigm is common in table question answering and fact verification tasks, where the model must integrate information across multiple rows and columns (Eisenschlos et al., 2020). While encoder-based

approaches benefit from the expressive capacity of pretrained language models, they are sensitive to input length constraints and serialization choices. As table size grows, truncation or heuristic selection of rows becomes necessary, limiting scalability.

### 3.1.1 Retrieval-Centered Approaches for Dense Tables

A fundamental limitation of applying large language models to tabular data arises from input length constraints. When tables are serialized into textual sequences, large tables frequently exceed the model's context window. As a result, many early approaches rely on truncation or heuristic row selection, which can discard relevant information and limit scalability in realistic settings.

To address this challenge, recent work has explored *retrieval-centered architectures* for table question answering and structured reasoning. In these systems, the LLM does not operate over the full table. Instead, retrieval mechanisms dynamically select relevant subsets of rows, columns, or cells before passing them to the model.

**Table-Aware Dense Retrieval**  Dense retrieval methods embed structured elements such as rows, cells, or columns into vector representations and retrieve the most relevant elements for a given query. Compared to standard document retrieval, table-aware retrieval must preserve structural relationships within the table, including column semantics and row-level dependencies.

Common indexing strategies include:

- **Row-level indexing:** Each table row is treated as a retrieval unit.

- **Cell-level indexing:** Individual cells are embedded together with their column context.

- **Column-aware embeddings:** Representations incorporate schema information to improve semantic alignment between queries and table attributes.

These representations enable scalable retrieval over large tabular datasets while maintaining awareness of structural relationships.

**Table Chunking and Indexing Strategies**  Another common strategy involves partitioning large tables into smaller chunks that can be processed independently. Typical chunking strategies include:

- **Row-based chunking**, where consecutive rows are grouped together,

- **Column-based chunking**, where only relevant columns are selected based on schema linking,

- **Hybrid chunking**, which combines row subsets with schema-aware filtering.

Chunked table segments can then be indexed using vector search or hybrid retrieval methods, allowing efficient identification of relevant subsets.

**Retrieval-Augmented Table Question Answering**  End-to-end retrieval-centered table QA pipelines typically follow three stages:

1. **Retrieval**: identify relevant rows, columns, or table segments.

2. **Context construction**: assemble retrieved elements into a serialized representation suitable for the LLM.

3. **LLM reasoning or translation**: generate answers, programs, or explanations based on the retrieved context.

This architecture parallels retrieval-augmented generation systems developed for text corpora, but must account for the structural dependencies that exist within tabular data.

**Interaction with LLM Context Limitations**   Large language models often struggle to reason over very large tabular contexts, particularly when numerical relationships span multiple rows or columns. Retrieval-centered pipelines mitigate this limitation by restricting the model input to the most relevant table segments. However, they introduce new challenges, including missed relevant rows, incomplete aggregation across retrieved chunks, and loss of global table structure.

Consequently, evaluation of retrieval-centered table QA systems often considers both final answer accuracy and retrieval effectiveness, reflecting the joint importance of retrieval quality and reasoning correctness.

**Retrieval-Augmented Table QA**   Retrieval-centered table QA pipelines typically follow three stages: (1) retrieval of relevant table segments, (2) construction of a compact serialized context, and (3) reasoning or program generation by the LLM. By restricting the input to relevant table subsets, these approaches improve scalability and reduce context overload.

However, retrieval introduces new challenges such as missed rows, incomplete aggregation across retrieved chunks, and loss of global table structure. As a result, evaluation often considers both answer accuracy and retrieval effectiveness.

## 3.2   LLMs as Reasoners over Structured Data

In the reasoning paradigm, LLMs are used to perform logical or numerical reasoning over structured representations, often without producing explicit executable programs. Methods in this category treat the model as a function mapping a structured state and a query directly to an answer (Chen et al., 2021; Zhu et al., 2021). Recent work has shown that prompting strategies such as chain-of-thought can improve multi-step reasoning over tables by encouraging intermediate computation steps (Wei et al., 2022).

Search-based reasoning strategies have also been explored for SQL generation. Alpha-SQL formulates query construction as a sequential decision process and uses Monte-Carlo Tree Search guided by LLM-generated actions to explore candidate SQL programs Li et al. (2025a).

Reasoning-based approaches are attractive due to their simplicity and flexibility, but they lack explicit grounding in execution environments. As a result, they may produce answers that are not faithful to the underlying data, particularly in tasks requiring precise aggregation or filtering. This limitation motivates hybrid approaches that combine reasoning with symbolic execution.

## 3.3   LLMs as Translators to Executable Programs

The translator paradigm maps natural language queries to executable programs such as SQL queries, data manipulation scripts, or domain-specific languages. Text-to-SQL systems represent the most extensively studied instance of this paradigm (Zhong et al., 2018; Yu et al., 2018b; Wang et al., 2020). In these systems, the correctness of the model output is determined by execution against a database engine, providing a clear semantic grounding.

Recent LLM-based approaches extend this paradigm beyond SQL to include data analysis code and transformation pipelines (Gao et al., 2023). Compared to reasoning-based methods, translator-based approaches offer stronger guarantees of faithfulness through execution. However, they require accurate schema linking and are sensitive to distribution shifts in database schemas and query structures.

### 3.3.1   Natural Language to SQL as a Canonical Translation Task

We examine natural language to SQL in greater depth because it represents the most execution-grounded and methodologically mature instance of the broader translator paradigm, making it a useful reference point for understanding design trade-offs that recur across structured data settings. The objective is to map a natural language query to an executable SQL program conditioned on a database schema, with correctness defined by execution results. Because it combines language understanding, schema reasoning, and formal semantics, NL-to-SQL has become a central benchmark for evaluating semantic parsing, cross-domain generalization, and execution-grounded learning. Recent surveys provide comprehensive coverage of model architectures,

datasets, and evaluation protocols in this space (Shi et al., 2025; Liu et al., 2025b). Table 3 situates representative methods within the broader translator paradigm.

Early neural approaches focused on sequence-to-sequence generation under constrained query structures. Models such as Seq2SQL, SQLNet, and TypeSQL were developed primarily for single-table datasets like WikiSQL, simplifying supervision by decomposing SQL generation into slot-filling or column-wise prediction (Zhong et al., 2018; Xu et al., 2017; Yu et al., 2018a). While effective in limited settings, these models struggled with compositional queries and multi-table schemas.

The introduction of the Spider benchmark marked a significant shift toward cross-domain evaluation with complex schemas and nested queries (Yu et al., 2018b). To address these challenges, schema-aware models explicitly represented database structure during encoding. RAT-SQL introduced relation-aware self-attention over schema graphs, enabling joint reasoning over questions and database elements (Wang et al., 2020). Subsequent work incorporated pretrained language models into this framework, improving both language understanding and schema linking while retaining explicit structural biases (Guo et al., 2019; Yu et al., 2021).

A complementary line of work focused on constrained and structure-aware decoding. Rather than relying solely on unconstrained sequence generation, models such as SmBoP reformulated SQL generation as bottom-up program construction, promoting compositional generalization (Rubin et al., 2021). PICARD introduced incremental constrained decoding, enforcing SQL grammar and execution constraints during generation without retraining the base language model (Scholak et al., 2021). These approaches reduce syntactic and semantic errors, particularly in complex query settings.

Beyond architectural innovations, recent work emphasizes decomposition, robustness, and learning from execution feedback. Methods such as RESDSQL and DIN-SQL decompose schema reasoning and query generation into subtasks, improving interpretability and generalization (Li et al., 2023a; Pourreza & Rafiei, 2023). Execution-guided decoding and execution-grounded learning incorporate feedback from database engines during inference or training, enabling models to optimize for execution correctness rather than exact query matching (Wang et al., 2018). Reinforcement learning–based approaches further leverage execution-based rewards to handle program equivalence and partial correctness.

The emergence of large pretrained language models has led to prompt-based and in-context learning approaches for NL-to-SQL. T5-based and PaLM-based models treat schemas as serialized text and rely on pretrained representations to perform translation with minimal task-specific architecture (Raffel et al., 2020; Chowdhery et al., 2023). Prompting-based methods, including GPT-style in-context decoding, demonstrate strong zero-shot and few-shot performance on simpler benchmarks, while execution filtering and validation are often required to ensure correctness on complex datasets.

More recent work explores agent-based NL-to-SQL systems that combine planning, tool use, and iterative execution. In these approaches, the language model decomposes a query into multiple steps, interacts with the database engine iteratively, and refines outputs based on execution feedback. Such systems are increasingly evaluated on enterprise-oriented benchmarks such as BIRD and Spider 2.0, which emphasize realistic schemas, noisy inputs, and multi-step analytical workflows (Li et al., 2023b; Lei et al., 2024).

Finally, unified evaluation frameworks such as UNITE and NL2SQL360 aim to standardize benchmarking across datasets, metrics, and execution settings, reflecting growing recognition that isolated accuracy numbers do not capture the full complexity of real-world text-to-SQL deployment (Lan et al., 2023; HKUSTDial, 2024).

Recent work has also begun to evaluate Text-to-SQL systems in more realistic conversational settings. MM-SQL introduces a benchmark for multi-turn Text-to-SQL interactions with diverse question types and conversational dependencies, highlighting challenges that are not captured by traditional single-query benchmarks Guo et al. (2025). Recent approaches also explore structure-guided prompting strategies to better align language model reasoning with database schemas. For example, SGU-SQL uses syntax-aware prompting and recursive decomposition to guide LLMs in incrementally constructing SQL queries Zhang et al. (2025).

Despite steady progress, NL-to-SQL remains challenging under realistic conditions. Performance remains sensitive to schema complexity, query nesting depth, and distribution shifts across domains. These limitations motivate continued exploration of hybrid approaches that integrate planning, retrieval, execution grounding, and agentic interaction. As such, NL-to-SQL continues to function not only as an application domain but also as a lens through which broader questions about language–structure interaction can be studied. Table 3 summarizes a representative set of natural language to SQL systems spanning early sequence-based models, schema-aware neural parsers, constrained decoding approaches, execution-grounded methods, and recent LLM-based prompting and agentic frameworks.

Table 3: Representative natural language to SQL methods

| METHOD | SCHEMA ENCODING | DECODING | EXECUTION USE | BENCHMARKS |
|---|---|---|---|---|
| Seq2SQL | Implicit column attention | Sequence-to-sequence | Execution-guided decoding | WikiSQL |
| SQLNet | Column-wise prediction | Slot filling | None | WikiSQL |
| TypeSQL | Type-aware encoding | Sequence-to-sequence | None | WikiSQL |
| X-SQL | Graph-based schema | Sequence-to-sequence | None | Spider |
| RAT-SQL | Relation-aware schema graph | Grammar-based decoding | None | Spider |
| RAT-SQL + BERT | Schema graph + pre-trained LM | Grammar-based decoding | None | Spider |
| GRAPPA | Pretrained relational encoder | Grammar-based decoding | None | Spider |
| SmBoP | Bottom-up operator tree | Program construction | None | Spider |
| PICARD | Pretrained LM + constraints | Incremental constrained decoding | Syntax and execution checks | Spider |
| RESDSQL | Schema reasoning decomposition | Grammar-based decoding | None | Spider |
| DIN-SQL | Decomposed subtask encoding | Sequence-to-sequence | None | Spider |
| DAIL-SQL | Prompted pretrained LM | In-context decoding | Execution filtering | Spider |
| Solid-SQL | Robust schema linking | In-context learning | Execution validation | Spider, BIRD |
| PaLM-based models | Serialized schema text | Sequence-to-sequence | Optional execution filtering | Spider |
| T5-based models | Serialized schema text | Sequence-to-sequence | Optional execution filtering | Spider, WikiSQL |
| ReSQL | Schema-aware encoder | Sequence generation | Execution-grounded learning | Spider |
| PaVeRL-SQL | Reinforcement learning | Hybrid decoding | Execution-based reward | Spider, BIRD |
| GPT-based prompting | Prompt serialization | In-context decoding | Execution filtering | Spider, CoSQL |
| Agent-based NL-to-SQL | Planner with tool calls | Stepwise generation | Iterative execution | Spider2.0, BIRD |
| Evaluation frameworks | Unified benchmark interfaces | Not applicable | Multiple metrics | UNITE, NL2SQL360 |
| Alpha-SQL | Schema serialization | MCTS-guided reasoning | Self-supervised SQL evaluation | BIRD |
| SGU-SQL | Structure-aware schema linking | Syntax-guided prompting | None | Spider |

Canonical benchmarks such as WikiSQL, Spider, BIRD, and Spider 2.0 progressively increase schema complexity and execution realism, shaping the evolution of NL-to-SQL methods listed in Table 3.

### 3.3.2 Beyond SQL: Structured Data Tasks without Explicit Query Languages

While natural language to SQL translation provides a canonical example of mapping language to executable programs, a substantial body of work applies large language models to structured data settings in which no explicit query language is available. In these tasks, the model operates over structured representations such as tables, graphs, or semi-structured records, but the output space consists of values, entities, explanations, or implicit actions rather than formally specified programs.

One prominent class of such tasks is table question answering and fact verification, where models are required to retrieve, aggregate, or compare values from tabular inputs. Benchmarks such as WikiTableQuestions and TabFact evaluate the ability to perform logical and compositional reasoning over tables without exposing an explicit execution interface (Pasupat & Liang, 2015; Chen et al., 2020). In these settings, execution is implicit: the model must internally simulate operations such as filtering, counting, or comparison, and correctness is assessed solely based on the final answer.

Related benchmarks such as TAT-QA and FinQA extend this paradigm to numerical and financial reasoning, requiring multi-step arithmetic over structured tables often combined with unstructured text (Zhu et al., 2021; Chen et al., 2021). These tasks further blur the boundary between reasoning and execution, as intermediate computations are neither explicitly represented nor directly verifiable, making faithfulness and error diagnosis challenging.

Knowledge graph question answering represents another structured data setting beyond SQL, where models operate over graph-structured data and produce entities or subgraphs as outputs. Datasets such as LC-QuAD and WebQuestionsSP evaluate the ability to interpret natural language queries over knowledge graphs, typically using SPARQL or logical forms as latent representations (Dubey et al., 2019; Yih et al., 2016). Although some approaches translate queries into formal graph query languages, many recent LLM-based methods rely on direct reasoning over serialized graph representations, again foregoing explicit execution guarantees.

Several recent frameworks explore deeper integration between LLMs and database systems. DB-GPT proposes an architecture where language models act as intelligent components for database interaction, supporting tasks such as query generation, prompt-driven optimization, and database-aware reasoning Zhou et al. (2024). Beyond relational databases, LLMs are increasingly applied to multimodal structured data settings. For example, TableGPT integrates table recognition with LLM reasoning to interpret and refine structured information extracted from table images Ren et al. (2025).

Finally, data transformation and manipulation tasks involve mapping natural language instructions to structured operations over tables or dataframes, often in the form of scripts or implicit transformations. Program-aided reasoning approaches generate intermediate code or function calls to support these tasks, but evaluation commonly focuses on end-state correctness rather than the fidelity of the generated operations (Gao et al., 2023).

Across these settings, the absence of explicit query languages introduces a common set of challenges. Without execution-grounded supervision, models may produce correct outputs for spurious reasons, masking underlying reasoning errors. At the same time, these tasks highlight the flexibility of language models as general-purpose interfaces to structured data, capable of operating in environments where formal query languages are unavailable, impractical, or too restrictive.

From the perspective of our taxonomy, these tasks occupy an intermediate position between reasoning-based and translator-based paradigms. They rely on structured representations and implicit operations, but lack the formal semantics and verifiability provided by explicit execution environments. This distinction has important implications for learning strategies, inference mechanisms, and evaluation protocols, which we examine in subsequent sections. Table 4 summarizes representative benchmarks for structured data tasks beyond NL-to-SQL, covering table question answering, numerical reasoning, and knowledge graph querying settings.

Table 4: Benchmarks for structured data tasks without explicit query languages

| BENCHMARK | DATA TYPE | METRIC | NOTES |
|---|---|---|---|
| WikiTableQuestions | Tables | Answer accuracy | Compositional table reasoning |
| TabFact | Tables | Binary accuracy | Fact verification |
| FeTaQA | Tables | ROUGE / F1 | Long-form table-to-text QA |
| TAT-QA | Tables + text | Exact match / numerical error | Hybrid numerical reasoning |
| FinQA | Financial tables | Numerical accuracy | Multi-step arithmetic reasoning |
| WebQuestionsSP | Knowledge graphs | Answer accuracy | SPARQL-style semantics |
| LC-QuAD 1.0 / 2.0 | Knowledge graphs | Answer accuracy | Complex graph queries |

These non-SQL tasks highlight structured data interaction scenarios that fall outside the scope of existing text-to-SQL surveys, yet exhibit many of the same modeling and evaluation challenges.

### 3.4 LLMs as Intermediate Computational Components

Within this abstraction, LLMs function as intermediate computational components rather than end-to-end executors. The model operates over representations of $S$ and $Q$, producing outputs that may require validation or execution by $\mathcal{E}$. This separation preserves the advantages of structured systems, including correctness guarantees and scalability, while enabling flexible natural language interaction.

Viewing LLMs through this interface perspective clarifies their role across different tasks. The same model may act as a translator from language to programs, a reasoner over serialized structures, or a planner that decomposes complex queries into executable steps. This perspective also highlights the importance of representation choices, execution grounding, and verification mechanisms, which are not captured when structured data tasks are treated solely as sequence prediction problems.

This abstraction provides the foundation for the taxonomy introduced in the next section, which categorizes existing approaches according to the functional role played by the LLM within structured data systems.

### 3.5 LLMs as Planners for Multi-Step Structured Tasks

Planner-based approaches use LLMs to decompose complex queries or analytical goals into sequences of intermediate steps, which may involve multiple queries, tool calls, or reasoning phases. In this setting, the model produces a plan rather than a single executable artifact (Zhou et al., 2023; Chen et al., 2023).

Planning is particularly relevant for tasks involving nested queries, data exploration, or iterative refinement. These methods often rely on external execution environments to carry out individual steps, with the LLM coordinating the overall process. While planning improves compositional generalization, it introduces additional complexity in control flow and error propagation.

### 3.6 LLMs as Agents Interacting with Structured Systems

Agent-based paradigms extend planning by allowing LLMs to interact iteratively with structured systems through observations and actions. In this setting, the model receives feedback from an execution environment and adapts subsequent actions accordingly (Yao et al., 2023; Schick et al., 2023). Structured data systems such as databases, APIs, or analytical tools serve as external components that the agent can query or manipulate.

Agent-based systems have also been proposed to support complex data analysis workflows. For instance, Data-Copilot introduces a code-centric data analysis agent that generates executable programs and reusable data interfaces to process and visualize large-scale datasets in response to user requests Zhang et al. (2023).

Agent-based approaches blur the boundary between reasoning and execution, enabling adaptive behavior in open-ended tasks. However, they raise challenges related to efficiency, reliability, and verification, as errors may accumulate over long interaction sequences.

### 3.7 Cross-Cutting Axes

Across these paradigms, two cross-cutting axes play a critical role in system design.

**Representation Strategy.** Methods differ in how structured data is represented for the LLM, ranging from naive linearization to schema-aware encodings and retrieval-based views (Shin et al., 2021). Representation choices affect both performance and scalability, particularly in settings with large schemas or databases.

**Learning Signal.** Approaches also vary in their supervision signals, including supervised learning on paired data, weak supervision via execution correctness, and in-context learning through prompting (Li et al., 2023a). Execution-grounded learning provides robustness but requires access to reliable execution environments during training or inference.

### 3.8 Applying the Role-Based Taxonomy in Hybrid Systems

While the taxonomy presented above categorizes systems according to the functional role played by the LLM, many modern structured data systems combine multiple roles within a single pipeline. For example, an analytical workflow may involve schema retrieval, reasoning over structured representations, generation of executable queries, and iterative interaction with external tools. To reduce ambiguity when mapping such hybrid pipelines to the proposed taxonomy, we introduce a small set of practical classification guidelines.

**Decision Rules for Classification**

- **Primary execution interface rule.** If the LLM produces an executable artifact whose correctness is determined through execution in an external environment (e.g., SQL queries or programs), the system is primarily categorized under the *translator paradigm*, even if intermediate reasoning or retrieval steps are used.

- **Implicit reasoning rule.** If the model directly maps a structured state and a query to an answer without generating an executable program, the system belongs to the *reasoning paradigm*, regardless of whether intermediate reasoning traces or chain-of-thought prompts are used.

- **Task decomposition rule.** If the LLM decomposes a complex query into a sequence of intermediate steps or subqueries before execution, the system is categorized under the *planner paradigm*, even when translation to SQL or code occurs within those steps.

- **Iterative interaction rule.** Systems that repeatedly interact with an execution environment through observations and actions are categorized under the *agent paradigm*, even when reasoning or translation occurs internally.

These rules emphasize the dominant operational role of the LLM during inference rather than auxiliary components that appear in the pipeline.

### 3.9 Summary

The taxonomy presented in this section organizes existing work according to the functional role of the LLM within structured data systems. This perspective reveals that many methods differ primarily in where the

boundary between language modeling and symbolic execution is drawn. The next section examines learning and inference mechanisms that support these paradigms, with an emphasis on prompting, finetuning, and tool integration.

## 4 Learning and Inference Mechanisms

This section examines the learning objectives and inference strategies used to adapt large language models to structured data tasks. While the functional role of the LLM determines how it interacts with structured systems, learning and inference mechanisms govern how effectively the model generalizes, grounds its outputs in execution semantics, and maintains faithfulness to the underlying data. Table 5 summarizes representative learning and inference mechanisms for large language models in structured data settings, highlighting differences in supervision, execution grounding, strengths, and limitations.

Table 5: Learning and inference mechanisms for structured data tasks

| MECHANISM | SUPERVISION | EXECUTION USE | STRENGTHS | LIMITATIONS |
|---|---|---|---|---|
| In-context prompting | None / few-shot | None | Flexible and data-efficient | Sensitive to schema size and prompt design |
| Supervised finetuning | Paired inputs and outputs | None | High task accuracy | Limited cross-domain generalization |
| Execution-guided decoding | Weak supervision | Validation during inference | Improves syntactic and semantic correctness | Requires reliable execution environment |
| Execution-grounded learning | Execution correctness | Training-time feedback | Robust to program equivalence | Higher training complexity |
| Tool-augmented inference | None or weak | External tools | Improves numerical and symbolic accuracy | Error propagation across steps |
| Agent-based interaction | Implicit or reinforcement | Iterative environment interaction | Supports multi-step reasoning | Efficiency and verification challenges |

### 4.1 Prompting and In-Context Learning

Prompting and in-context learning provide a lightweight mechanism for applying pretrained LLMs to structured data tasks without parameter updates. In this setting, structured inputs such as tables or database schemas are serialized into text and provided as part of the input context, often accompanied by task demonstrations (Brown et al., 2020). For structured reasoning tasks, prompting strategies such as chain-of-thought encourage the model to produce intermediate reasoning steps, improving performance on multi-step computations (Wei et al., 2022).

In NL-to-SQL and related translation tasks, prompt-based approaches enable rapid adaptation to new schemas but remain sensitive to input length constraints and schema complexity. Empirical studies show that few-shot prompting can achieve competitive accuracy on simpler benchmarks but degrades on complex, cross-domain datasets such as Spider, particularly for queries involving joins and nested subqueries (Yu et al., 2018b). These limitations motivate more structured learning approaches.

### 4.2 Supervised Finetuning on Structured Corpora

Supervised finetuning remains a dominant approach for high-performance structured data systems. Models are trained on paired inputs and outputs, such as natural language questions and SQL queries or table-question-answer pairs. Finetuned models based on pretrained architectures such as T5 and BERT demon-

strate strong gains across multiple benchmarks when sufficient task-specific data is available (Raffel et al., 2020; Herzig et al., 2020).

In NL-to-SQL, finetuning on curated datasets improves schema linking and syntactic correctness but may lead to overfitting to benchmark-specific query patterns. Cross-domain evaluation on unseen schemas reveals persistent generalization gaps, highlighting the tension between data efficiency and robustness (Yu et al., 2018b). These observations motivate the use of auxiliary supervision signals beyond exact query matching.

### 4.3 Execution-Grounded Learning and Decoding

Execution grounding incorporates feedback from structured execution environments into training or inference. Execution-guided decoding filters or reranks candidate outputs based on syntactic validity or execution results, improving correctness without modifying model parameters (Wang et al., 2018). Constraint-based decoding methods such as PICARD further enforce grammar and type constraints during generation, reducing invalid outputs in NL-to-SQL tasks (Scholak et al., 2021).

More recent approaches integrate execution signals into the learning objective through reinforcement learning or weak supervision. By optimizing for execution correctness rather than exact program match, these methods account for program equivalence and reduce sensitivity to annotation artifacts (Li et al., 2023a). Execution-grounded learning has proven particularly effective in structured settings where formal semantics are available.

### 4.4 Tool Use and Program-Aided Inference

Program-aided inference frameworks augment LLMs with external tools such as interpreters, database engines, or analytical libraries. In this paradigm, the model generates intermediate programs or function calls that are executed to obtain intermediate results, which are then incorporated into subsequent reasoning steps (Gao et al., 2023). This approach separates high-level reasoning from low-level computation, improving numerical accuracy and scalability.

Tool-augmented inference is closely related to agent-based paradigms, where the LLM interacts iteratively with an execution environment. While these methods increase flexibility, they introduce challenges related to efficiency, error propagation, and verification, particularly in multi-step structured tasks.

### 4.5 Discussion

Across structured data tasks, learning and inference mechanisms reflect a trade-off between flexibility and grounding. Prompting-based methods emphasize adaptability but lack robustness guarantees, while finetuning and execution-grounded approaches improve correctness at the cost of task-specific supervision. Understanding these trade-offs is essential for designing reliable LLM-based structured data systems.

The next section examines how these mechanisms are evaluated and discusses limitations of existing benchmarks and metrics in structured data settings.

## 5 Evaluation and Limitations

Evaluation plays a central role in assessing the effectiveness of large language models for structured data tasks. Unlike unstructured text generation, structured data settings provide well-defined semantics and executable environments, enabling objective measures of correctness. At the same time, existing benchmarks and metrics capture only a subset of the challenges encountered in realistic structured data interaction.

Table 6: Benchmark landscape for LLMs as interfaces to structured data. We group datasets by modality/task and include both canonical benchmarks and robustness-oriented variants commonly used to stress schema linking and distribution shift.

| DATASET / SUITE | TASK TYPE | METRIC (TYP.) | NOTES / WHAT IT STRESSES |
|---|---|---|---|
| **NL2SQL: single-turn and conversational** | | | |
| WikiSQL (Zhong et al., 2018) | NL-to-SQL (single table) | Execution accuracy | Early large-scale benchmark; limited compositionality. |
| Spider (Yu et al., 2018b) | NL-to-SQL (cross-domain) | Execution accuracy | Multi-table, cross-domain generalization; schema linking sensitivity. |
| SParC (Yu et al., 2019b) | Conversational NL-to-SQL | Execution accuracy | Context carryover and interaction history; multi-turn error propagation. |
| CoSQL (Yu et al., 2019a) | Conversational NL-to-SQL | Execution accuracy | Dialogue phenomena, clarification, and conversational ambiguity. |
| **NL2SQL: enterprise realism and evaluation frameworks** | | | |
| BIRD (Li et al., 2023b) | Enterprise NL-to-SQL | Execution accuracy | Large-scale DB-grounded evaluation; BI/analytics realism; harder schemas. |
| Spider 2.0 (Lei et al., 2024) | Enterprise workflows (tool + DB) | Task success / execution | End-to-end enterprise workflows (often multi-step); deployment-style difficulty. |
| UNITE (Lan et al., 2023) | Unified NL2SQL evaluation suite | Execution accuracy | Aggregates datasets for systematic comparison across NL2SQL settings. |
| NL2SQL360 (HKUSTDial, 2024) | Multi-angle evaluation framework | Multiple metrics | Evaluation harness covering robustness and broader diagnostic views. |
| LogicCat Liu et al. (2025a) | Reasoning-intensive NL-to-SQL | Execution accuracy | Chain-of-thought SQL benchmark with arithmetic, physics, and hypothetical reasoning. |
| TINYSQL Harrasse et al. (2025) | NL-to-SQL with controllable complexity | Execution accuracy | Structured benchmark designed to gradually increase SQL query complexity. |
| NL2SQL-Bugs Liu et al. (2025c) | NL-to-SQL error analysis | Semantic correctness / execution accuracy | Benchmark targeting semantic SQL errors where queries execute but fail to match the intended user meaning. |
| SynSQL-2.5M Li et al. (2025b) | Large-scale synthetic NL-to-SQL | Execution accuracy | Large synthetic dataset containing millions of NL–SQL pairs for training large models and studying generalization at scale. |
| **NL2SQL: robustness / perturbation (schema-linking stress tests)** | | | |
| Spider-Realistic (Yu et al., 2021) | NL-to-SQL robustness | Execution accuracy | Removes explicit column mentions to test realistic user phrasing and schema alignment. |
| Spider-DK (Gan et al., 2021) | NL-to-SQL robustness | Execution accuracy | Tests domain knowledge / schema grounding under distribution shift. |
| **Table QA / table fact verification** | | | |
| WikiTableQuestions (Pasupat & Liang, 2015) | Table QA | Answer accuracy | Compositional reasoning over semi-structured tables. |
| TabFact (Chen et al., 2020) | Table fact verification | Accuracy (binary) | Faithfulness in tabular entailment/verification setting. |

Table 6 – *Continued from previous page*

| DATASET / SUITE | TASK TYPE | METRIC (TYP.) | NOTES / WHAT IT STRESSES |
|---|---|---|---|
| FeTaQA (Nan et al., 2022) | Long-form table QA | Text metrics + human eval | Free-form answers grounded in tables; stresses generation faithfulness. |
| **Table numerical / hybrid table-text reasoning** | | | |
| FinQA (Chen et al., 2021) | Numerical reasoning (finance) | Numeric accuracy | Multi-step arithmetic and program-like reasoning over financial tables. |
| TAT-QA (Zhu et al., 2021) | Hybrid table+text QA | EM / numeric error | Hybrid evidence across tables and text; arithmetic + span reasoning. |
| **Knowledge graph QA (KGQA)** | | | |
| WebQuestionsSP (Yih et al., 2016) | KGQA (KB semantic parsing) | Answer/F1 | Canonical KBQA dataset with SPARQL/logical-form grounding. |
| LC-QuAD 1.0 (Trivedi et al., 2017) | KGQA (Wikidata/DBpedia) | Answer accuracy | Large QA set for KG querying; query templates + KG grounding. |
| LC-QuAD 2.0 (Dubey et al., 2019) | KGQA (complex) | Answer accuracy | More complex questions and broader coverage than LC-QuAD 1.0. |

Table 6 summarizes representative benchmarks used to evaluate large language models as interfaces to structured data. The table spans canonical natural language to SQL datasets, enterprise-oriented and workflow-based benchmarks, robustness and perturbation suites, as well as table reasoning and knowledge graph question answering tasks. Together, these benchmarks reflect increasing schema complexity, interaction depth, and deployment realism, and illustrate how evaluation practices have evolved beyond isolated query accuracy toward more realistic structured data settings.

## 5.1 Benchmarks and Metrics

Early benchmarks such as WikiSQL emphasize single-table query generation with limited compositional complexity, enabling rapid progress but encouraging models that exploit dataset-specific patterns (Zhong et al., 2018). More recent benchmarks, including Spider, SParC, and CoSQL, introduce cross-domain databases, multi-table schemas, and conversational settings, shifting the focus toward schema generalization and contextual reasoning (Yu et al., 2018b; 2019b;a).

For table-based reasoning tasks, datasets such as TAT-QA and FinQA evaluate numerical and logical reasoning over tabular inputs (Zhu et al., 2021; Chen et al., 2021). These benchmarks typically use exact-match accuracy or numerical error metrics, which do not capture reasoning faithfulness or robustness to perturbations. Knowledge graph benchmarks such as LC-QuAD assess query answering over graph-structured data, but often rely on templated queries that limit linguistic diversity (Dubey et al., 2019).

Execution-based evaluation, commonly used in NL-to-SQL tasks, provides a principled notion of correctness by comparing query results rather than surface forms. However, execution accuracy alone does not reveal whether a model's reasoning process is faithful to the underlying data or whether errors are masked by coincidental equivalence.

**Emerging evaluation datasets.** Recent work has also introduced new datasets designed to probe limitations of current Text-to-SQL systems beyond standard benchmarks such as Spider and BIRD. LogicCat Liu et al. (2025a) focuses on reasoning-intensive SQL generation, including arithmetic, physics-based reasoning, and hypothetical scenarios, and contains 4,038 questions paired with over 12,000 reasoning steps across multiple databases. TINYSQL Harrasse et al. (2025) provides a controllable benchmark with gradually increasing SQL complexity, enabling systematic analysis of model behavior across difficulty levels. NL2SQL-Bugs Liu et al. (2025c) targets semantic correctness by evaluating cases where syntactically valid SQL queries fail to capture the intended semantics of the natural language query. Large-scale synthetic datasets such as

SynSQL-2.5M Li et al. (2025b) further explore data diversity and scale, providing millions of automatically generated NL–SQL pairs for training and generalization studies.

## 5.2 Generalization and Robustness

A recurring limitation across benchmarks is sensitivity to schema complexity and distribution shift. Models that perform well on in-domain databases often exhibit substantial performance degradation when evaluated on unseen schemas or domains (Yu et al., 2018b). Similar issues arise in table reasoning benchmarks when column names, value distributions, or table layouts differ from those observed during training.

Prompt-based LLM approaches further amplify these challenges, as performance depends on serialization choices, prompt length constraints, and demonstration selection. These factors complicate fair comparison across methods and limit reproducibility.

## 5.3 Faithfulness and Verification

Faithfulness remains a central challenge for LLM-based structured data systems. Reasoning-based approaches may produce fluent but incorrect outputs that are inconsistent with the underlying data. Translator- and execution-grounded approaches mitigate this issue but are not immune to subtle errors arising from incorrect schema linking, incomplete constraints, or partial execution feedback.

Current benchmarks rarely evaluate intermediate reasoning steps or require explicit justification aligned with execution traces. As a result, models may achieve high accuracy without developing reliable internal representations of structured semantics.

As illustrated in Figure 3, errors may arise during schema linking, logical formulation of aggregations or joins, execution against underspecified programs, or post-processing and explanation of results. Importantly, these errors can compound across stages, producing fluent but incorrect outputs that are difficult to detect using final-answer accuracy alone.

These failure modes highlight the limitations of evaluation protocols that rely solely on execution accuracy, motivating the need for verification mechanisms that expose intermediate decisions and execution traces (Figure 3).

## 5.4 Summary of Evaluation Challenges

Overall, existing evaluation protocols emphasize task-specific accuracy while underrepresenting broader concerns such as robustness, faithfulness, and interaction complexity. Addressing these limitations will require benchmarks that integrate diverse structured data modalities, support multi-step interaction, and incorporate evaluation criteria beyond final-output correctness.

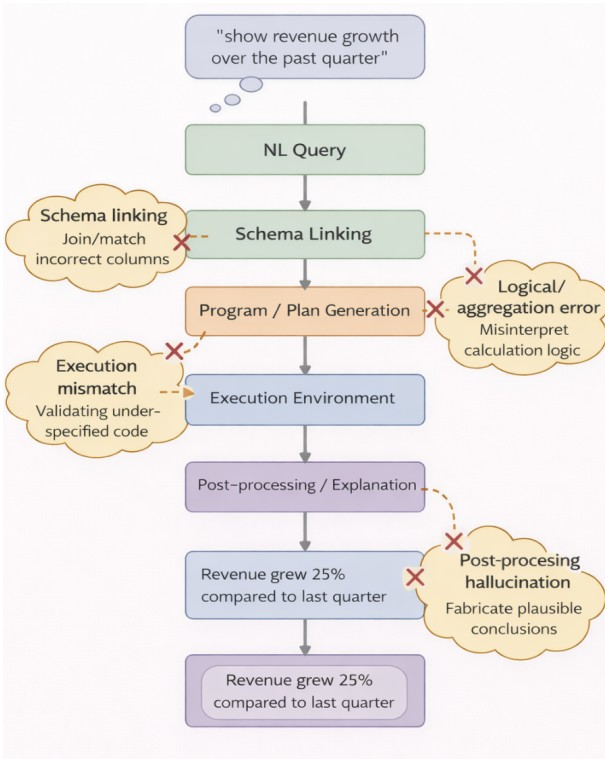

Figure 3: Failure modes in large language model interaction with structured data systems.

# 6 Applications

While much of the literature on large language models for structured data focuses on benchmark tasks and modeling techniques, practical deployments increasingly reveal system-level considerations that are not captured by standard evaluations. This section surveys representative application domains in which LLMs act as interfaces to structured data systems in practice. For each domain, we highlight recurring system patterns as well as characteristic failure modes that motivate the methodological and evaluation challenges discussed in earlier sections.

## 6.1 Data Analytics and Business Intelligence

Large language models are increasingly deployed as natural language interfaces for data analytics and business intelligence systems. In this setting, the model mediates between user queries expressed in natural language and structured analytical backends such as data warehouses or OLAP engines. Typical system architectures combine schema retrieval, query translation, execution feedback, and result summarization, allowing users to perform ad hoc analysis, generate dashboards, or explore data without direct interaction with SQL or visualization tools.

Text-to-SQL models form the core of many such systems, with recent work extending beyond single-query translation toward multi-step analytical workflows and enterprise-scale databases (Yu et al., 2018b; Wang et al., 2020; Scholak et al., 2021; Li et al., 2023b; Lei et al., 2024). Execution-guided decoding and constrained generation are commonly employed to reduce syntactic and semantic errors, while agent-based approaches increasingly support iterative refinement and follow-up queries (Yao et al., 2023).

A dominant failure mode in analytics settings arises from schema ambiguity and aggregation mismatch. Models may select semantically plausible but incorrect columns, misinterpret temporal constraints, or gen-

erate queries that satisfy syntactic requirements while violating domain-specific business semantics. Because such queries often execute successfully, errors may remain undetected, leading to misleading analytical results. These challenges highlight the limits of execution accuracy as a sole evaluation criterion and motivate stronger verification and semantic alignment mechanisms.

## 6.2 Enterprise Workflows: Governance, Semantics, and Permissions

Enterprise deployments impose additional constraints that fundamentally shape how LLMs interact with structured data. Beyond schema complexity, real-world databases are governed by access controls, data ownership policies, and organizational definitions of metrics and entities. In these environments, LLMs typically operate within controlled execution pipelines that enforce row-level permissions, restrict schema visibility, and validate generated queries against governance rules.

Recent enterprise-oriented benchmarks such as BIRD and Spider 2.0 reflect these constraints by introducing realistic schemas, noisy metadata, and multi-step analytical tasks (Li et al., 2023b; Lei et al., 2024). Systems designed for these settings often integrate schema filtering, permission-aware retrieval, and post-generation validation layers to ensure compliance. Rather than generating unrestricted queries, the model acts as a planner whose outputs are mediated by policy-aware execution components.

Failure modes in enterprise workflows frequently stem from misalignment between linguistic intent and organizational semantics. Even when schema linking is technically correct, models may combine metrics or tables in ways that violate implicit business rules or governance constraints. Such errors are not easily detected through execution alone and underscore the need for explicit semantic verification and policy-aware evaluation beyond traditional benchmark settings.

## 6.3 AutoML and Feature Engineering over Structured Stores

Large language models are also increasingly applied to automate feature engineering and pipeline construction over structured datasets. In this paradigm, the model generates data transformations, aggregations, or modeling pipelines expressed as code or declarative operations, which are executed by external systems such as AutoML frameworks or data processing engines. The LLM functions as a high-level planner or generator, while optimization and execution are delegated to specialized components.

Program-aided reasoning approaches exemplify this pattern by separating symbolic computation from natural language reasoning, allowing models to invoke interpreters or libraries for numerical operations and data manipulation (Gao et al., 2023). Related work explores decomposition-based planning and tool-augmented inference to support complex transformation pipelines (Zhou et al., 2023; Chen et al., 2023).

A characteristic failure mode in this setting is spurious feature construction. Generated transformations may inadvertently encode data leakage, violate causal assumptions, or introduce distribution shifts that degrade generalization. Because such pipelines often execute successfully and may even improve short-term metrics, errors can be difficult to diagnose without domain-aware validation and careful evaluation protocols.

## 6.4 Scientific Structured Data

Scientific domains such as materials science, biology, and clinical research rely heavily on structured datasets, including experimental tables, registries, and curated knowledge bases. Large language models are increasingly used to query, summarize, and integrate these resources, often in conjunction with unstructured text such as publications or laboratory notes. In this setting, structured data interaction frequently spans heterogeneous sources with varying schemas and levels of curation.

Knowledge graph question answering and table-based numerical reasoning benchmarks provide partial abstractions of these tasks, capturing aspects of entity-centric querying and multi-step computation (Hogan et al., 2021; Yih et al., 2016; Dubey et al., 2019; Zhu et al., 2021; Chen et al., 2021). However, real-world scientific workflows often involve implicit assumptions about units, experimental conditions, and data provenance that are not explicitly encoded in schemas.

Failure modes in scientific applications are particularly consequential. Incorrect aggregation, unit mismatches, or silent schema drift can lead to invalid scientific conclusions, even when outputs appear linguistically plausible. Because ground truth is often unavailable at scale, verification remains a major challenge, emphasizing the need for execution grounding, provenance tracking, and domain-specific validation mechanisms in LLM-based structured data systems.

### 6.5 Summary

Across application domains, large language models serve as interfaces rather than replacements for structured data systems. While they enable flexible and accessible interaction, practical deployments reveal recurring challenges related to semantic alignment, verification, and governance. These observations motivate the evaluation criteria and future research directions discussed in the following section.

## 7 Open Problems

Beyond consolidating existing work, the abstraction and taxonomy introduced in this survey surface several open research problems that are not well captured by task-centric formulations. We highlight a small set of concrete problems that cut across structured data modalities and interaction paradigms.

**Faithful Reasoning under Implicit Execution.** In structured data tasks without explicit execution environments, such as table reasoning and knowledge graph question answering, large language models often produce correct outputs for spurious reasons. How can models expose verifiable intermediate reasoning steps or latent execution traces without sacrificing the flexibility that makes implicit reasoning attractive?

**Schema-Robust Generalization.** Despite strong performance on benchmark datasets, many LLM-based systems remain brittle under schema variation, renaming, or reorganization. Developing representation and learning strategies that generalize across unseen structured schemas remains an open challenge central to real-world deployment.

**Unified Evaluation across Structured Modalities.** Current benchmarks and metrics are siloed by task and data modality, limiting comparability across approaches. Can evaluation frameworks be designed to assess faithfulness, robustness, and interaction complexity across tables, databases, and knowledge graphs under a shared set of principles?

**Control and Verification in Agentic Structured Systems.** Agent-based interaction with structured systems enables powerful multi-step analytics but introduces new failure modes related to planning, tool misuse, and error accumulation. How can agentic LLM systems be constrained, verified, or audited when interacting iteratively with structured execution environments?

## 8 Discussion and Future Directions

This survey examined the role of large language models as interfaces to structured data systems through a unified abstraction and a role-based taxonomy. By organizing prior work according to the functional role played by the model—encoding, reasoning, translation, planning, and agentic interaction—we highlighted common design patterns and recurring challenges that span traditionally separate task domains.

A central theme emerging from this analysis is the importance of execution grounding. Tasks with explicit execution environments, such as natural language to SQL translation, benefit from objective correctness criteria and verifiable outcomes. In contrast, structured data tasks without formal query languages rely on implicit execution and internal reasoning, making faithfulness, robustness, and error diagnosis substantially more difficult. Bridging this gap remains a key challenge for future research.

Several open directions follow from this perspective. First, there is a need for benchmarks that more faithfully reflect real-world structured data interaction, including complex schemas, noisy inputs, and multi-step ana-

lytical workflows. Such benchmarks should support evaluation beyond final-output accuracy, incorporating criteria related to reasoning trace alignment, robustness to schema variation, and interaction efficiency.

Second, learning paradigms that combine the flexibility of prompting with the reliability of execution-grounded supervision warrant further exploration. Hybrid approaches that integrate symbolic constraints, tool feedback, or lightweight program induction may offer a path toward improved generalization without sacrificing interpretability.

Third, agent-based interaction with structured systems introduces new opportunities and risks. While iterative tool use enables complex analytical behavior, it also raises challenges related to efficiency, verification, and safety. Developing principled abstractions for agent control and evaluation will be critical as such systems move toward deployment.

Finally, the abstraction and taxonomy introduced in this survey suggest opportunities for unifying research across structured data modalities. Rather than treating tasks such as table reasoning, database querying, and knowledge graph interaction as isolated problems, future work may benefit from shared representations, learning objectives, and evaluation frameworks that span these settings.

In summary, large language models have the potential to fundamentally reshape how users interact with structured data. Realizing this potential will require advances not only in model architectures, but also in execution grounding, evaluation methodology, and system-level design. We hope this survey provides a foundation for such efforts and supports the development of more reliable and general structured data interfaces.

**How to use this survey.** This survey is intended as a design and analysis reference rather than a static catalog of methods. Researchers can use the proposed taxonomy to identify which functional role—encoding, reasoning, translation, planning, or agentic interaction—is most appropriate for a given structured data task, and to reason about trade-offs between execution grounding, flexibility, and robustness. Practitioners can use the benchmark and failure mode analyses to anticipate common sources of error when deploying LLM-based structured data interfaces, particularly in enterprise and scientific settings. Finally, the abstraction introduced in Section 2 provides a common language for comparing approaches across traditionally siloed domains such as text-to-SQL, table reasoning, and knowledge graph question answering.

**Limitations.** This survey prioritizes conceptual synthesis and abstraction over exhaustive cataloging of all existing systems, and as a result some application-specific implementations are discussed only at a high level. We also focus primarily on textual interaction with structured data, leaving deeper treatment of multimodal structured data (e.g., images, time series, or sensor streams) to future work.

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
