# OpenReview forum: "Large Language Models as Interfaces to Structured Data: A Survey"
_TMLR — Rejected by TMLR_

### Review · Reviewer_3E1v · 2026-02-22

**Summary Of Contributions:**

Please summarize the contributions of the paper in your own words. Please also list any key strengths and/or weaknesses, but please be mindful that this is NOT a substitute for the next two text boxes.

The survey defines LLMs as structured data interfaces and presents a unified formulation that captures tasks based on the structured state, query/input, output space, and execution environment.
It suggests a role-based taxonomy that groups previous research based on what the LLM does (encoding, reasoning, translation/planning, and agentic execution) in relation to structured data such as tables, relational databases, and knowledge graphs. It then moves on to the topic of evaluation and challenges, specifically robustness, faithfulness/verification, and scalability.

**Strengths:**

**1) Clear unifying abstraction:** The paper’s (S,Q,O,E) formulation is a strong organizing device: it cleanly separates the structured state, the user/control input, the output space, and the execution environment, and it’s reused consistently across paradigms

**2) Role-based taxonomy is intuitive and readable:** Organizing the functional role of the LLM by cataloging it to encoding, reasoning, translation, planning, agents is a coherent perspective of the paradigm this tackles, which helps compare papers across task families.

**3) It acknowledges cross-cutting design axes:** Calling out representation strategy and learning signal as cross-cutting axes is helpful, and it already gestures at retrieval-based views as part of representation choices.

**Weaknesses:**

The survey identifies truncation/heuristic row selection as the scalability bottleneck, but it does not substantially synthesize retrieval-first approaches for large tables, beyond briefly noting retrieval-based representations.

**The paper flags the scalability problem clearly:  “While encoder-based approaches are sensitive to input length constraints and serialization choices. As table size grows, truncation or heuristic selection of rows becomes necessary, limiting scalability.”**

Because the paper highlights that large tables trigger truncation or heuristic row selection, a natural next step is to treat structured data retrieval as the primary optimization target for dense tables. Instead of trying to squeeze more of the table into the model input, the focus shifts to building retrieval and indexing that can reliably surface the right rows, columns, or table regions, so the model only sees what it needs. This is becoming an interesting direction precisely because it sidesteps context window limits through engineering and modeling choices like table aware embeddings, chunking strategies, and retrieval time aggregation, rather than relying on longer prompts.

**2)  Strong problem framing and taxonomy, but it stops short of being operational guidance for practitioners.**
The paper explicitly positions itself as conceptual rather than a practical “methods you can apply” guide. It says it “prioritizes conceptual synthesis and abstraction over exhaustive cataloging of all existing systems,” and that “some application-specific implementations are discussed only at a high level.” It also frames the contribution “as a design and analysis reference rather than a static catalog of methods.”

That choice is valid, but it creates a real limitation. If researchers are trying to build or evaluate an LLM structured-data system, they will get strong language for describing the space, but fewer concrete takeaways like a decision checklist, recommended baselines per setting, minimum reporting standards, or step-by-step evaluation protocols. The survey’s own statements about staying at a high level are exactly why this gap exists.

**3)  Difficulty of role-based taxonomy application in real-world systems.**
One weakness the paper does not really acknowledge is that its role based taxonomy can be hard to apply consistently to real systems, because many modern pipelines are hybrids and the boundaries between roles are fuzzy. The survey says it “categorize[s] prior work according to the functional role played by the LLM” but it also notes that “agent-based approaches blur the boundary between reasoning and execution,” and that “many methods differ primarily in where the boundary between language modeling and symbolic execution is drawn.”

Given that, what would be an ideal way to address this taxonomy on systems that do not rely on one underlying tool? Tightening the taxonomy with concrete mapping criteria and examples of borderline cases would make the framework more reproducible and more useful for comparing systems.

**Audience:**

Yes

**Audience Explanation:**

Yes, structured querying using LLMs is an emerging field in machine learning, the survey synthesizes many of the prominent studies made in this area to define their proposed taxonomies

**Broader Impact Concerns:**

Given that this study is a systematic review of LLM querying on structured data, there would not be any need in adding a broader impact statement.

**Claims And Evidence:**

Yes

**Claims Explanation:**

The paper is generally convincing because it clearly states its aim to provide “a unified conceptual framework” and an “abstract formulation” that organizes existing methods. It then grounds its organization in prior literature by presenting “a taxonomy of existing approaches” and explicitly “categorize prior work according to the functional role” the model plays in structured data pipelines. This role-based taxonomy is presented as a unifying lens that supports systematic comparison across paradigms, which supports the main claims in a clear and coherent way.

**Requested Changes:**

**Clarify taxonomy application with concrete examples and borderline case.**

The role-based taxonomy is a useful organizing lens, but it would benefit from clearer mapping guidance for systems that span multiple roles (for example, retrieval + generation + execution checking, or agentic systems where reasoning and execution are interleaved). Adding a short set of decision rules plus 3 to 5 worked examples of “borderline” pipelines would reduce ambiguity, improve reproducibility of categorization, and make the taxonomy more actionable for readers who want to place their own systems within the framework.


**Add coverage of dense table retrieval and retrieval-centered table QA for long, dense structured data.**

The paper already notes that large tables can require truncation or heuristic row selection due to input length constraints, which limits scalability in realistic settings. A concise subsection on retrieval-centered approaches would strengthen the survey’s coverage of how the community handles dense, long-form tables when the full table cannot be fit into the model context. This could include (a) table-aware dense retrieval, (b) indexing and chunking strategies for tables, and (c) end-to-end retrieval-augmented table QA pipelines and how they are evaluated (d) efficacy of these methods with interfacing with Large Language Models which have a hard time parsing large context tables for data analysis and querying.

---

> ### Author Response · Authors · 2026-03-16
> **Review Response - added sections 3.1.1 and 3.8**
>
> Thank you for the thoughtful and constructive feedback. We appreciate the positive assessment of the paper’s conceptual framework, particularly the unified $(S,Q,O,E)$ formulation and the role-based taxonomy for organizing LLM interactions with structured data.
>
> We have revised the manuscript to address the reviewer’s suggestions regarding taxonomy clarity and scalability considerations for dense tabular data.
>
> 1. Clarifying taxonomy application in hybrid systems
>
> We agree that modern structured data pipelines often combine multiple roles (e.g., retrieval, reasoning, SQL generation, and execution feedback), which can make categorization under the role-based taxonomy less straightforward. To improve clarity and reproducibility, we added a new subsection:
>
> Section 3.8: “Applying the Role-Based Taxonomy in Hybrid Systems.”
>
> This section introduces a set of decision rules for categorizing systems based on the dominant functional role played by the LLM during inference. These guidelines are intended to help readers consistently place hybrid systems within the proposed taxonomy even when multiple components are involved in the pipeline.
>
> 2. Coverage of dense table retrieval and retrieval-centered table QA
>
> We also agree that scalability for large tables is an important issue. While the original text discussed truncation and heuristic row selection as limitations of encoder-style approaches, it did not synthesize retrieval-centered solutions.
>
> To address this, we added a new subsection:
>
> Section 3.1.1: “Retrieval-Centered Approaches for Dense Tables.”
>
> This subsection discusses:
>
> Table-aware dense retrieval methods for indexing rows, cells, and schema elements
>
> Chunking and indexing strategies for large tables
>
> Retrieval-augmented table QA pipelines and how retrieved subsets are integrated into LLM reasoning or program generation
>
> The role of retrieval in mitigating LLM context limitations when processing dense tabular data
>
> These additions strengthen the discussion of scalability and provide a clearer picture of how modern systems handle dense structured datasets.

---

### Review · Reviewer_ypn2 · 2026-03-07

**Summary Of Contributions:**

This paper surveys LLMs as interfaces to structured data and positions its contribution around a unified abstraction of structured tasks, plus a role-based taxonomy in which LLMs act as encoders, reasoners, translators, planners, or agents. It also covers learning/inference mechanisms, evaluation challenges, application domains, and open problems.

**Audience:**

Yes

**Audience Explanation:**

LLMs are increasingly used to interact with structured data, and a well-executed survey that unifies existing approaches would be useful to the community.

**Claims And Evidence:**

No

**Claims Explanation:**

I am not yet convinced that the paper supports its claims as a current and comprehensive survey.

The survey’s literature coverage appears outdated. Given the pace of recent work in this area, the paper’s citations largely include papers from 2023 and earlier, which makes it difficult to view the manuscript as a current or comprehensive survey.

While the paper aims to unify LLM-based approaches to structured data broadly, the technical coverage still appears to be centered primarily on SQL.

The paper seems to have figures that contain incorrect information. Most significantly, figure 2 includes two section 3s, and links to irrelevant tables/figures.

The references need substantial revision, there are published works where the arxiv version was cited, there are multiple papers with duplicated citations (ex. Raffel 2020a, 2020b), and most importantly there are citations that deviates from existing work which I list below:

- *Wrong author name and paper name:* X. Li et al. Resdsql: Decoupling schema and question representation for text-to-sql. arXiv preprint, 2023b.

- *Paper name does not match*: M. Pourreza et al. Din-sql: Decomposed in-context learning of natural language to sql via iterative reasoning. arXiv preprint arXiv:2304.11015, 2023.

- *Wrong first author*: Wenhu Chen et al. Finqa: A dataset of numerical reasoning over financial data. In Conference on Empirical Methods in Natural Language Processing (EMNLP), 2021.

- *Wrong author list*: Wenhu Chen, Hongmin Wang, Jianshu Chen, Yunkai Zhang, Shiyu Wang, Yiming Zhang, Kun Wang, et al. Tabfact: A large-scale dataset for table-based fact verification. In International Conference on Learning Representations (ICLR), 2020.

**Requested Changes:**

- Expand the survey to cover more recent work.

- Broaden the coverage beyond SQL-centric settings, or narrow the paper’s claims accordingly.

- Correct figure and cross-reference errors.

- Thoroughly revise the bibliography for accuracy, completeness, and consistency.

---

> ### Author Response · Authors · 2026-03-16
> **Response to Review**
>
> Thank you for the detailed and constructive feedback. We appreciate the reviewer’s comments regarding literature coverage, scope, figure consistency, and reference accuracy. We have revised the manuscript to address these concerns.
>
> 1. Expanding coverage of recent work
>
> We agree that it is important for the survey to reflect recent developments in this rapidly evolving area. The manuscript has been updated to include recent technical work from 2024–2025, including systems such as OmniSQL, TableGPT, and related structured-data LLM interfaces. These works are now discussed in the relevant subsections of Section 3, and have been added to the summary tables (Tables 3 and 5) and referenced in Section 5 where appropriate. This ensures that the survey reflects current research trends in LLM-based structured data interaction.
>
> 2. Clarifying the scope beyond SQL-centric systems
>
> While natural language–to–SQL systems represent a large portion of existing research, we agree that the survey should clarify its scope beyond SQL-centric settings. In the manuscript, we explicitly note at the beginning of the natural language–to–SQL subsection that this area is discussed in greater detail because it is currently the most mature and well-studied paradigm for LLM interaction with structured data.
>
> At the same time, the survey framework is designed to cover broader structured data modalities including tables, relational databases, and knowledge graphs.
>
> 3. Correcting Figure 2
>
> Thank you for pointing out the issues with Figure 2. We have improved this figure by removing redundant sections and cross-refs.
>
> 4. Revising the bibliography
>
> We performed a thorough revision of the bibliography to improve accuracy and consistency. Specifically, we:
>
> Corrected author names and paper titles where mismatches occurred.
>
> Updated citations to reference published conference versions when available instead of earlier arXiv versions.
>
> Removed duplicate citations.
>
> We thank the reviewer again for these suggestions, which helped improve the accuracy and completeness of the survey.

---

### Review · Reviewer_Ykuy · 2026-03-15

**Summary Of Contributions:**

This paper reviews different works leveraging LLMs as an interface to deal with structured data. Compared to existing surveys, which may focus on individual structured data types including tables and graphs, this work unifies different structured data into one framework.

**Audience:**

Yes

**Audience Explanation:**

This survey discusses how LLMs are being leveraged to deal with structured data. This would interest readers working on various domains, including the ones working on AI and ML and the ones working on specific application domains.

**Claims And Evidence:**

Yes

**Claims Explanation:**

The main claims include the insufficiencies of the existing surveys and how existing LLM-based structured data processing pipelines work. Most of the claims look convincing.

**Requested Changes:**

1. The expression could be more rigorous. For example, it is claimed that 'This survey presents a theory-oriented overview of LLMs for structured data', but there does not seem to be any theoretical component.

2. I would recommend to highlight the benefit of unifying different conceptual frameworks. For example, LLM-based approaches for processing table data and graph data are largely different and used in different domains, what is the advantages of unifying these two distinct approaches in one framework?

3. The covered technical works should be updated. Currently, all 2025 works are survey works. No technical works in 2025 are reviewed.

4. Figure 2 could be improved. the part corresponding to Section 3.1.1 seems to be blurred.

---

> ### Author Response · Authors · 2026-03-16
> **Response to Reviewer**
>
> Thank you for the helpful feedback and for the positive assessment of the paper’s motivation and relevance to the TMLR audience. We have revised the manuscript to address the reviewer’s suggestions regarding terminology, the motivation for unifying structured data paradigms, coverage of recent work, and figure clarity.
>
> 1. Clarifying the benefit of unifying structured data paradigms
>
> We agree that the benefit of unifying different structured data domains (e.g., tables, relational databases, and knowledge graphs) should be stated more explicitly.
>
> To address this, we added a paragraph in the introduction, which mentions the advantages of the unified abstraction.
>
> - Reveal common design patterns across systems that operate on different structured data types
>
> - Provide a shared abstraction for describing LLM-based structured data pipelines
>
> - Enable systematic comparison across domains that have traditionally been studied separately (e.g., table QA vs. graph querying)
>
> - Highlight cross-cutting architectural choices, such as representation strategies and execution grounding
>
>
>
> 2. Updating coverage of recent technical works
>
> We agree that coverage of recent technical work is important. The manuscript has been updated to include recent 2025 technical systems, including examples such as OmniSQL TableGPT, NL2SQL Bugs, which are now discussed in the relevant subsections of Section 3 and 5.
>
> These works have also been added to the summary tables (Tables 3 and 5) and referenced in Section 5, ensuring that the survey reflects the most recent developments in LLM-based structured data processing.
>
> 3. Improving Figure 2 clarity
>
> Thank you for pointing out the issue with Figure 2. We have updated this figure for better clarity and accuracy.

---

### Decision · Action_Editor_ymbp · 2026-04-19

**Recommendation:** Reject

**Audience:**

Yes

**Audience Explanation:**

LLMs are increasingly used to interact with structured data, and a well-executed survey that unifies existing approaches would be useful to the community.

**Claims And Evidence:**

No

**Claims Explanation:**

This survey paper received mixed reviews, but I concur with Reviewer ypn2 that it is not ready for publication. The main reasons are:
- The papers covered are not sufficiently extensive. Only SQL-related sections are detailed, while many sections discuss only a handful of works from more than 3 years ago (for example, Section 4.4 discusses tool use for LLMs but cites only PaL from 2023, even though entire surveys on the topic have been published since then [1]).
- There are hallucinated citations. This is unacceptable for any paper, but especially a survey paper, where the point is to give an accurate representation of the field. Hallucinated references call into question a survey paper's credibility and value.

[1] Qu, Changle, et al. "Tool learning with large language models: A survey." Frontiers of Computer Science 19.8 (2025): 198343.